# Celebrities and Medical Awareness—The Case of Celine Dion and Stiff-Person Syndrome

**DOI:** 10.3390/ijerph20031936

**Published:** 2023-01-20

**Authors:** Abdulrahman Elsalti, Mohammad Darkhabani, Mohamad Aosama Alrifaai, Naim Mahroum

**Affiliations:** International School of Medicine, Istanbul Medipol University, Istanbul 34083, Turkey

**Keywords:** stiff-person syndrome, public awareness, Celine Dion

## Abstract

The positive role of celebrities in spreading important medical information and contributing to increasing public awareness regarding the diagnosis, treatment, and prevention of various medical conditions cannot be overemphasized. Interestingly and importantly at the same time, this impact is not related to the rarity of the disease, as very rare diseases are looked up by the public due to the fact that a celebrity suffers from this disorder. Therefore, if taken seriously and used to address the public in regard to critical medical conditions, such as screening for cancer or the importance of vaccines in fighting infections, celebrities could have a huge impact in this field. As previously shown in the medical literature, the recent announcement of the famous Canadian singer Celine Dion concerning her newly diagnosed stiff-person syndrome has influenced the public interest regarding the syndrome which manifested as an increased search volume related to the disorder as seen in Google Trends. In brief, in this short communication we aimed to address the phenomenon of celebrities’ impact on public apprehension, revise the syndrome for the medical community, and emphasize taking advantage of such involvement of celebrities for improving the spread of highly important medical information for the public.

## 1. Introduction

On 9 December 2022, the Canadian singer Celine Dion posted a video declaring her new diagnosis of “stiff person syndrome”. The singer explained that years of suffering from various symptoms, particularly spasms including her vocal cords, have come to an end when she was lately diagnosed with stiff-person syndrome. While addressing the audience in tears, Dion announced the cancellation of her upcoming singing tours as she needs to focus on treatment, hoping that she could recover from the syndrome in a short period. Following the announcement, Google Trends (GT) has showed an increase interest of the public regarding the syndrome. The interest was coupled, in terms of search, with the name of the singer. In fact, this was not the first time, once a celebrity is affected with a medical issue or disease, either common or rare, the specific condition or disease becomes a trend. If such events are taken seriously, a huge benefit could result, in the form of an increased awareness of the public in all aspects, including disease prevention, diagnosis, and treatment. Moreover, in relation to stiff-person syndrome, as the case of Celine Dion, refreshing the knowledge of medical personnel is of great importance as well.

Hereby, we aimed in our short paper to illustrate the positive influence of celebrities in the medical field. We focused on the role of celebrities with regard to awareness concerning medical conditions, even rare ones, alongside various aspects of diagnosis, treatment, as well as prevention. Stiff-person syndrome, as a rare disorder, was also presented in conjunction with the new and latest therapeutic options.

## 2. The Role of Celebrities and Influencers in Raising Medical Awareness

Throughout history, celebrities have always been looked at as the top of the pyramid in the societal hierarchy. With the introduction of social media in the past decade, people have noticed the impact that influencers and celebrities can have on the general public. Such examples can be related to diseases and their therapies as well.

The most recent news regarding the diagnosis of the famous Canadian singer, Celine Dion, with stiff-person syndrome was definitely a shock to many of her fans, many of whom had no idea about the syndrome in the first place. In fact, it is not uncommon for celebrities to disclose conditions they suffer from publicly, and on many occasions such announcements could contribute to increased public awareness regarding common and rare disorders alike [1].

One of the most iconic figures of all time, Mohammad Ali, an American boxer, was diagnosed with Parkinson’s disease in 1984. Despite the fact that not much was known about the disease among the public in the 1980s, the diagnosis of Parkinson’s disease in the most prominent figure in sports history spiked the public interest considerably. Ali also played such a huge role in raising awareness about the disease. This included advocating for more funding for both research and therapy, donating huge amounts of money for these purposes, to a point where Ali even contributed to new steps in the management of disease, such as starting physical exercise early after the diagnosis of Parkinson’s disease, which can help improve the patient’s quality of life and reduce symptoms [2]. Similarly, in relation to Parkinson’s disease, the well-known comedy actor Michael J. Fox and the foundation in his name have contributed generously to researching Parkinson’s disease, particularly regarding treatment [3]. The developments achieved in this domain have made the clinical trials for new treatment options possible and applicable.

In 2013, another respected celebrity and actress Angelina Jolie announced to her fans that she carries an inherited BRCA1 mutation. The actress declared publicly as well that she underwent a bilateral risk-reducing mastectomy because of the documented mutation. Subsequently, studies reveal that this event singlehandedly led to a 36% increase in BRCA testing in the US [4]. Some researchers went further to term the incident as the “Angelina Jolie Effect”, further proving the role of celebrities in increasing public awareness related to medical conditions, particularly in terms of prevention. Needless to say, this topic is critical when to comes to the prevention and early detection of malignant diseases.

Moreover, recent examples of celebrities and medical awareness include Emilia Clarke suffering two brain hemorrhages while working on the show “Game of Thrones”. After her brain injuries, she founded “Same You”, a charity to help to develop better neurorehabilitation care for people who have experienced brain injuries [5]. Another example is Selena Gomez and her diagnosis with systemic lupus erythematous (SLE). The celebrity revealed her diagnosis with SLE on a live TV show in 2015, talking about her struggles with SLE during an interview. Two years later, Gomez announced on her Instagram page that she had to undergo a renal transplant secondary to SLE. Gomez also shared some websites with her followers so that they can educate themselves on SLE, and its complications and treatment. Interestingly, following the announcement and the developments of the condition of Gomez, the effect was reported in one study showing an increase in Google searches on the term “what is lupus” after Gomez shared her condition with the public [6].

In fact, the list just keeps going, as earlier this year, the popstar Justin Bieber announced on his Instagram page that he suffers from the rare Ramsay Hunt syndrome. Bieber explained that the rare syndrome led to the facial paralysis seen on one side of his face. Subsequently, as expected, the public declaration generated a huge surge of empathy, solidarity, and awareness from his devoted followers and fans [7].

## 3. What Is Stiff-Person Syndrome?

Stiff-person syndrome (SPS), previously known as “stiff man syndrome”, is a rare autoimmune neurological disorder with a prevalence of 1 case per million people. Similar to most autoimmune diseases, a higher incidence in women is noted. Most of the affected patients are diagnosed between the ages of 20 and 50 years old. The mechanisms related to the pathogenesis of SPS are described mainly by the decreased inhibition of the central nervous system (CNS). This is seemingly caused by the autoantibodies directed against the glutamic acid decarboxylase (GAD), which is a rate-limiting enzyme in the synthesis of gamma-aminobutyric acid (GABA), a major inhibitory neurotransmitter in the CNS [8]. Although GAD-specific T-cell clones have been isolated from the cerebrospinal fluid (CSF) of patients with SPS, the role of T cells in the pathogenesis of SPS remains uncertain. Furthermore, as GAD is highly expressed in pancreatic beta-cells, 30% of patients with SPS have type 1 diabetes mellitus (DM) among other associated conditions, including thyroiditis and vitiligo [9].

Clinically, SPS has been subdivided into three categories. These are classic SPS, partial SPS, and paraneoplastic SPS [10]. The classical form of SPS is seen in the majority of affected patients, around 70% to 80%. The classical type is characterized by stiffness and rigidity that usually involve the trunk as the first place and then progress slowly to affect the proximal muscles of the limbs. Patients with SPS walk with a tendency to fall with a wide-based and unsteady posture. Long-term effects can be as severe as fixed spinal deformities, such as lumbar or cervical lordosis, or even sudden death caused by diaphragmatic spasm. In turn, the partial form of SPS is characterized by the involvement of only one extremity, usually a lower limb, sparing the trunk, hence the term “partial”. Finally, paraneoplastic SPS has a similar presentation as the classical form except for negative antibodies, the anti-GAD antibodies. Nevertheless, the presence of anti-amphiphysin antibodies has been extensively reported with this form of the disease [11]. Moreover, anti-amphiphysin was suspected to play a pathogenetic role.

In regard to the diagnosis of SPS, it is generally confirmed by electromyography together with testing for the GAD antibodies following the documentation of classical clinical findings, such as axial muscle stiffness and painful spasms. In patients with SPS, the electromyography shows continuous motor activity of both agonist and antagonist muscle. An amount of 10,000 IU/mL is needed as a cut-off value for GAD antibodies to be considered positive [12].

The main aims in the treatment plan of patients with SPS are to improve the function and mobility of the patients by controlling the symptoms with an emphasis on the spasms. With reference to spasms, benzodiazepines, especially diazepam, are frequently used as the initial drug of choice with the dose gradually increased over weeks in order to reach an optimal range of the dose [13]. Patients resistant to initial treatment or with unsatisfactory response to treatment can be treated with a GABA receptor agonist. Usually, the drug baclofen is used as an add-on therapy [14]. IVIG, B-cell depletion, and plasma exchange are considered in patients who are unresponsive or intolerant to benzodiazepines. Recently, in a small cohort of 10 patients with stiff-person syndrome in the UK, autologous stem cell transplantation in SPS patients demonstrated significantly good results following transplantation [15]. Patients enrolled in the UK cohort had severe SPS and previously failed conventional treatment with immunosuppressive agents. After the procedure, patients improved in terms of symptoms, and most importantly they stopped immunosuppressive therapy.

Eventually, of paramount importance, a gluten-free diet showed great results in patients with stiff-person syndrome. In a study including 20 patients with SPS, the patients were recommended to adopt a gluten-free diet. Of the 20 SPS patients enrolled, 12 patients improved on the diet, while in another 7 patients the gluten-free diet was the only treatment given [16]. Therefore, being a key figure, gluten was proposed to play a role in the pathogenesis of the disease. Taking into consideration the feasibility of this approach, a gluten-free diet could aid patients suffering from SPS.

## 4. Google Trends as a Tool of Measuring the Impact of Celebrities

From the history of celebrities and their influence on medical awareness, we can safely say that Celine Dion will probably have a similar impact on public interest after her diagnosis with stiff-person syndrome. This impact can already be seen, as the unfortunate news announced by the celebrity raised attention of stiff-person syndrome among the general public, indeed. This public interest was captured by Google Trends (GT). GT is a free tool provided by Google LLC. through a free access website which can be utilized by everyone to explore and track events or news trending worldwide [17]. In the GT website, by adding the search term or topic the internet surfer is interested in, one can determine where and when to focus the search. For instance, worldwide or country specific, or in the last day or month. In our case, stiff-person syndrome and Celine Dion were used as search terms, whereas the hours following the announcement were determined as the time selected. Importantly, in GT, the data presented have the features of anonymity, categorization, normalization, aggregation, and of being a representative sample of the general population. The beneficial potential that GT has in capturing public attention and perhaps providing important flags prior to events occurring has been previously described [18].

As in the case of Celine Dion and stiff-person syndrome, the increased attention that the celebrity gained has become trending, as an increased number of people searched for the terms “stiff-person syndrome” and “Celine Dion” in the hours following the disclosure as presented in (Figure 1).

Following the occasion, which was started by Celine Dion addressing the public live, declaring the new diagnosis with stiff-person syndrome, the worldwide public interest had shifted toward searching for the rare medical entity, symptoms, diagnosis, and treatment. Interestingly, what might be unintentionally neglected can become the center of focus in a short while. For example, undiagnosed people with SPS may seek medical care, or people might search for donations supporting SPS patients and research in this field.

## 5. Conclusions

The role and influence of celebrities in spreading awareness in terms of medical problems as well as preventive medicine are of great importance. Though rare, the difficulty in diagnosing stiff-person syndrome might become easier following the announcement of its diagnosis by a celebrity, as a result of the introduction of the syndrome to a broader audience. This would contribute to the earlier detection and intervention in addition to providing opportunities for research aiming to find directed treatment options. Taking advantage of their impact, celebrities should be encouraged to collaborate with the medical community when it comes to the early diagnosis, treatment, and prevention of diseases.

## Figures and Tables

**Figure 1 ijerph-20-01936-f001:**
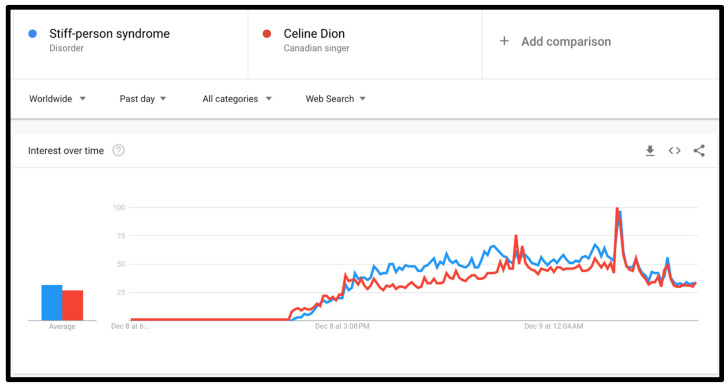
GT search volume related to the search terms “Stiff-person syndrome” and “Celine Dion” on the day the announcement was made by the singer.

## Data Availability

Not applicable.

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
