# Peer review of "Celebrities and Medical Awareness—The Case of Celine Dion and Stiff-Person Syndrome"

_ijerph, 2023, doi:10.3390/ijerph20031936_

Round 1

Reviewer 1 Report

This appeared an interesting report of a rather predictable phenomena - the search for a disease etc on google when a particular celebrity has reported having the condition.   The authors have already published one such report in 2018 although in that case it was not a "rare" dissease but a rare cause of death in a well known disease, rheumatoid arthritis.

Thus the novelty here relies on the very rare stiff person syndrome and a well known celebrity Celine Dion (even my partner knew of this singer!).

The graph shows clearly that there were searches for each that peaked on the day of the report of her illness.  

It would be interesting to see how the graph(s) have changed in the month following the public disclosure.  Has there been any longer-term interest in SPS?

2. What evidence is there, from other conditions, that recognition of a disorder has improved or is likely to improve following this rather short-lived peak or interest?

It is worth reporting but the English needs minor improvements and the text is repetitive and could be reduced - the message is clear enough.

The probably dont need to cite three of their own papers, and should cite more of others on the same Google Trends approach.

Author Response

Dear Reviewer,

Thank you for your important comments.

We edited our paper according to your notes as follows:

  1. Unfortunately, we could not add an additional graph from GT, as the results are confusing, owing to the fact the Celine Dion as a "search term" was recently searched extensively following protests by her fans for not choosing her among one of the greatest singers.
  2. Regarding the evidence of effects of such interests, we mentioned the Angelina Jolie effects in terms of prevention, as well as the contribution to research in the fields mentioned.
  3. We omitted one of our references, we kept the RA case as well as the Plague outbreak owing to their importance and implications.

Thank you again.

The authors. 

Reviewer 2 Report

This communication is well written and focusses on the impact that celebrities may have on specific diseases if they are diagnosed with the disease. Perhaps another example that should eb included is Michael J Fox and Parkinson's disease, in that the MJ Fox foundation has funded a lot of research in this disease.

Regarding the section on SPS, paraneoplastic SPS which is excitingly rare is usually associated with anti-amphyphysin antibodies, so the comments about this group being seronegative needs to be changed. Aside of benzodiazepines other antispasmodics are also used such as baclofen, dantrolene etc and these can be first line treatments. Autologous stem cell transplantation can be very effective in this disease. Finally the authors need to mention that a large number of patients with SPS have gluten sensitivity and that in those cases gluten free diet may have a therapeutic effect.

Author Response

Dear Reviewer,

Thank you so much for your valuable comments.

We edited our paper according to your notes as follows:

  1. Michael J Fox - was mentioned alongside the contribution the foundation has. A reference was added.
  2.  Anti-amphiphysin antibodies were addressed.
  3. Autologous stem cell transplantation as a treatment option was added with a proper reference.
  4. Similarly, gluten-free diets was described as well.

Thank you again,

The authors.

Round 2

Reviewer 1 Report

This has not changed except for additions.  It is not more concise and there is much about SPS where the recent Dalakas review (10) would be more appropriate than trying to reference a selection of individual articles, some also review-type, others early primary. 

The highlighted additions are not relevant and should be omitted (and the new references just added at the end) . 

There are still three self-references, 1, 14, 15. 

THe English really does need some small edits.

Author Response

Dear Reviewer,

Thanks for your comments.

As you might not have seen our new cover letter to the editor, we could not edit the references in the manuscript provided here. That's why, you see the newly added references and the omitted one (15), highlighted and at the end as such.

In addition, the "irrelevant" additions, as you called it, where recommended by the second reviewer. These were regarding, anti-amphiphysin antibodies, stem cell transplantation, and gluten free diet.

Moreover, based on the editor's office request, the manuscript was extended to 2000 words (instead of around 1500 in the original paper). So we re-edited the entire text, extended it and sent back by an email. Resubmitted here again.

Hope our answer is clear and meets your expectation.

Thank you,

The authors.